# 2D Personality of Multifunctional Carbon Nitrides towards Enhanced Catalytic Performance in Energy Storage and Remediation

Gajendra Kumar Inwati [1,*], Virendra Kumar Yadav [2], Ismat H. Ali [3], Sai Bhargava Vuggili [2,4], Shakti Devi Kakodiya [5], Mitesh K. Solanki [6], Krishna Kumar Yadav [7], Yongtae Ahn [8], Shalini Yadav [9], Saiful Islam [10] and Byong-Hun Jeon [8,*]

1   Department of Chemistry, HVHP Institute of Post Graduate Studies and Research, Sarva Vishwavidyalaya, Kadi 382715, India
2   Department of Microbiology, School of Sciences, P P Savani University, Kosamba, Surat 394125, India; yadava94@gmail.com (V.K.Y.); saibhargavavuggilli@gmail.com (S.B.V.)
3   Department of Chemistry, College of Science, King Khalid University, Abha 61413, Saudi Arabia; ihali@kku.edu.sa
4   School of Nano Sciences, Central University of Gujarat, Gandhinagar 382030, India
5   Department of Bioscience, Rani Durgavati University, Jabalpur 482001, India; shaktikakodiya1310@gmail.com
6   School of Engineering and Applied Sciences, Ahmedabad University, Ahmedabad 380009, India; mitesh4physics@gmail.com
7   Faculty of Science and Technology, Madhyanchal Professional University, Ratibad, Bhopal 462044, India; envirokrishna@gmail.com
8   Department of Earth Resources & Environmental Engineering, Hanyang University, 222-Wangsimni-ro, Seongdong-gu, Seoul 04763, Korea; ytahn83@hanyang.ac.kr
9   Department of Civil Engineering, Rabindranath Tagore University, Raisen 464993, India; shaliniy2000@gmail.com
10  Civil Engineering Department, College of Engineering, King Khalid University, Abha 61421, Saudi Arabia; sfakrul@kku.edu.sa
*   Correspondence: gajendrainwati@gmail.com (G.K.I.); bhjeon@hanyang.ac.kr (B.-H.J.)

**Abstract:** Numerous scholars in the scientific and management areas have been overly focused on contemporary breakthroughs in two-dimensional objects for multiple prospective applications. Photochemical and electrocatalytic functions of integrated circuits associated with multi-component tools have been enhanced by designing the macro- and microstructures of the building blocks. Therefore, the current research attempts to explore a larger spectrum of layered graphitic carbon nitrides (g-C$_3$N$_4$) and their derivatives as an efficient catalyst. By executing systematic manufacturing, optimization, and evaluation of its relevance towards astonishing energy storage devices, adsorption chemistry, and remediation, many researchers have focused on the coupling of such 2D carbon nitrides combined with suitable elementals. Hybrid carbon nitrides have been promoted as reliable 2D combinations for the enhanced electrophotocatalytic functionalities, proved by experimental observations and research outputs. By appreciating the modified structural, surface, and physicochemical characteristics of the carbon nitrides, we aim to report a systematic overview of the g-C$_3$N$_4$ materials for the application of energy storages and environments. It has altered energy band gap, thermal stability, remarkable dimensional texturing, and electrochemistry, and therefore detailed studies are highlighted by discussing the chemical architectures and atomic alternation of g-C$_3$N$_4$ (2D) structures.

**Keywords:** carbon allotropes; 2D atomic structures; surface alteration; energy storage; environmental concerns

## 1. Introduction

Over the last several years, rapid innovations in the field of innovations and research have enabled to upgrade the effectiveness and development of humankind, while the latest

developments in modern scientific research have raised global survival and human life. Nevertheless, due to their accurate handling and functionality in numerous purposes such as photonic, pharmaceutical, and environmental implications, multi-component instruments, or equipment, are becoming an essential aspect [1,2]. Notably, as every operation requires energy in some form or another, constant demand in energy storage and conversion has emerged as the most promising situation. Natural resources have been utilized from the beginning of time, and as a result they are depleting owing to unrestricted power consumption and use. Numerous experts in this perspective are attempting to develop an alternative supply to handle such energy-related challenges [3,4]. They have continued to work on multiphase sensors and systems, such as superconductors, capacitors, and solar systems, in addition to storing and transforming photon energy into electrical energy [5–7]. Across many sectors, the use of such electronics to gradually shift in individual parts has been encouraged, and it has been demonstrated to be fairly successful in addressing such energy-related issues. Major advances in nanostructured materials and their promising effects have sparked interest among researchers in the professional and technical worlds. In this logic, the intended paragraph aims to cover a broader class of two-dimensional materials, particularly graphitic carbon nitride (g-$C_3N_4$) [2,8,9]. The composite group of carbon nitrides loaded with appropriate materials has been emphasized by performing a comprehensive construction, generalization, and exploration of its importance in spectacular superconductors in recent years. By taking into account practical results, several researchers tried to modify the carbon-based doped 2D substances to moderate their physicochemical features, and reported their outstanding catalytic outputs [10–12]. In the present environment, multimetallic components of such hybrid substances are being discovered by targeting carbon nitride materials, and a significant method is developed for manufacturing, characterization, and supercapacitor studies in material sciences and their applications [9,12]. It is well addressed that the excellent energy gap, physical and thermodynamic strength, and chemical resistance of these smart 2D materials have already been recognized as a potential technology for power storage, and thus lots of research is carried out in this particular field of materials sciences. Various physicochemical operations were used to modify the chemical constituents and surface finish in atomically layered materials of carbon nitrides for a spectrum of uses [13,14]. The discrete quality of surface, opacity, and catalytic aspects for supercapacitor and environmental applications make them desired candidates among the 2D morphologies in carbon-based materials [8,15]. While the metal and metal-doped semiconducting nanomaterials and their applications were developed by modifying the crystal band structures and surface alteration [16,17], these advanced nano-range materials were found to be very reliable and efficient towards improved performance of macro- and microsystems in modern techniques and apparatus.

In other ways, a slew of new industries and companies have sprung up to develop and produce advanced technologies, which are utilized to make fundamental building components for humankind and everyday life. Because of the growing need for nanocomposites and related ingredients, chemical experiments and enterprises are springing up all over the place. Hazardous residues, co-products, and intermediates, on the other hand, are produced from a variety of industries and laboratories, posing a serious risk to human health as well as the environment [18,19]. Because of their extended lifecycle, toxic effects, and erosion, organic and heavier metal-based building materials are also highly toxic wastes. As a result, they have piqued the interest of scientists and inventors to use innovative equipment and technologies to assess environmental challenges including water, air, and soil qualities cleanup. The studies have revealed several ways to address these ecological consequences, including the development of heterostructures for photocatalytic activity, sewage disposal, and antimicrobial uses [20,21]. Terminated trash from many sources is the primary factor in the spread of different symptoms, which has made human existence more problematic and dangerous in recent years. As a result, people have been able to chronicle the people's development in improving material design and adaptation. Multiple categories of substances are being produced at the microscopic and nanometer size

using nanostructures and nanoscience ideas. The use of nanometer-sized components as predicted analogs or mediators has improved the efficiency of catalytic performance in complex mixtures. The architectural, photonic, and environmental characteristics of this kind of transformed nanomaterials are considered valuable in the production of nanoelectronics, therapeutic instruments, and atmospheric nanosensors [6,7,22]. Alkali, transitional, and radioactive isotopes, including heterocyclic organic molecules, are among the heterostructures whose manufacture and design have been aided by major progress in physical science and research. Diverse elemental constituents in the form of multimetallic nanostructures have been reported and researched for their morphological, spectroscopic, and chemical capabilities. Semiconductors such as $CdO$, $ZnO$, $Fe_3O_4$, $MnO_3$, $TiO_2$, $SnO_2$, and $MgO$ have been used to increase physicochemical qualities [23,24]. These materials are generally utilized and valued for their biological, microelectronics, and aerospace applications. In comparison to bulk, their small dimensions and greater surface regions have enabled nanoparticles to act as one of the most desirable catalytic alternatives. Doping principles might be used to modify the compositional and electronic structure of these kinds of nanocomposites; hence, a lot of studies have been carried out in this field [25–27]. In contrast, additional kinds of conjugated polymers have grown rapidly and evolved in the domain of blended composite materials. Countless hybrid series of such 2D nanostructures, such as carbon-based 2D graphene, reduced graphene-oxides, CNT, carbon nitrides ($C_3N_4$), and $MoS_2$, have been found and applied to resolve the energy storage and environmental applications [18,28,29]. In this article, a systematic overview of the 2D carbon nitrides and their derivatives is addressed, including sequential manufacturing, characterizations, uses in energy conversion, and environmental impacts, by discussing their characteristic optical, structural, and surface properties.

## 2. Developments and Engineering of g-$C_3N_4$

The g-$C_3N_4$ is a two-dimensional substance having increased nitrogen appropriate to a given active surface area, similar to graphene. These positions are regarded as sensitive regions of the $C_3N_4$ framework due to the electrostatic exposure, which aids in catalytic enhancement via photogenerated electrons transmission. Novel chemical approaches could potentially be used to change the photocatalytic efficiency at specific parameters. Due to its good physicochemical robustness and inexpensive price, it has been classified as a soft polymer semicrystalline framework [9,12]. The creation of advanced textiles and their distinctive behavior were investigated in order to construct macro- to nanodevices at a greater scale. Blend compounds are created by mixing some different factors in their natural and altered chemical forms. Carbonaceous and inorganic compounds have been utilized extensively in the creation of a new type of matter, particularly for energy storage technologies and other applications. These materials have been modified to attain the desired qualities for diverse applications, increasing the number of researcher methodologies and pathways to construct these 2D materials [30]. In the field of 2D materials, polymeric g-$C_3N_4$ has emerged as the primary focus among the carbon-containing layered structures. The energy bands of CB and VB for the g-$C_3N_4$ were studied around 2.7 eV, which is a favorable region in visible light. The catalytic activity of such semiconducting materials fundamentally depends upon the band gap values, and thus the g-$C_3N_4$ have an optimal energy which could be altered for the multiple uses in a simple way. The lightweight, flexible character of the g-$C_3N_4$ possesses strong mechanical strength and chemical stabilities in order to use as efficient catalysts for biomedical, optoelectronics, and catalysis uses. Furthermore, g-$C_3N_4$ has the capacity to endure temperature, strong acid, and basic media. As compared to other hybrid photocatalysts, g-$C_3N_4$ can be readily prepared by easy thermal polycondensing using simple N-rich precursors, for example, dicyanamide, cyanamide, melamine, melamine cyanurate, and urea. These cost-effective synthesis and physicochemical properties of g-$C_3N_4$ make it a futuristic material for multiple applications [12,31].

The characteristics of the g-$C_3N_4$ structures and functionalities could be regulated by determining the electrical, physicochemical, and refractive functions, due to that the non-

crystalline to crystalline form and varied compositions affect the catalytic performances. Therefore, the manufacturing process must be moderated by changing the experimental setup during synthesis [32]. Thermolytic fusion of precursor salt, such as melamine ($C_3N_3(NH_2)_3$), cyanamide, and dicyandiamide ($C_2N_4H_4$, DCDA), is a common novel approach to creating gCN compounds associated with Liebig's melon. The ribbon-like features are created by connected networks of heptazine ($C_6N_7$) subunits, and the production technique produces polymer composites with a limiting constitution around that of Liebig's melon. The layers of $C_3N_3$ (s-triazine) units are linked via $sp^2$-bonded N atoms. In 1982, the composition of the original component "cyamelurine", also known as tri-s-triazine ($C_6N_7H_3$), which contains this heptazine strong unit, was first discovered [33]. Komatsu [34] looked into Liebig's melon and hydromelonate compounds, which he thought could be successors to a substantially graphitic g-$C_3N_4$ substance, and assumed it would be made up of sheets of heptazine units joined by trigonal N atoms. Later, Bojdys et al. used DCDA to produce crystalline 2D materials in a molten salt (eutectic LiCl–KCl) solvent solution. Proportions of $C_6N_8.5H1.5Li0.8Cl0.2$ were discovered using a mix of analysis approaches. Extensive XRD patterns were discovered in the space group of P6$_3$cm, which indicates an interlayer ($d_{002}$) with 3.36-inch spacing.

To assign high-quality catalysts, many approaches for the synthesis of g-$C_3N_4$ and structural modification have been used. The higher-temperature-induced polycondensation process was usually performed in conjunction with calculations at a set time. Purified and polymerized g-$C_3N_4$ and nitrogen-rich chemicals with flaming qualities such as urea, melamine, thiourea, and uranic acid are frequently made using this low-cost approach [9,35]. Thermal treatment in a furnace at a temperature is favored by a community of researchers for sequentially preparing g-C3N4, while physical and chemical methods for surface modifications are carried out later. The outcome is often yellow with compacted solid structures. Changes in temperature and acidic media, as well as open situations, have been used to modify microstructures. The superconductive semiconductor g-$C_3N_4$ has been investigated as a substance that can be transformed by changing experimental variables. These counterparts are nitrogen-rich molecules that include reactive groups. To obtain the desired qualities, band structures and surface modifications were made. Aside from that, for the fabrication and engineering of polymer nanocomposites or nanosheets such as graphene or CNTs, a variety of synthesis processes have been used [22,36]. The PVD, CVD, and spin coating processes are particularly suited for fabricating these thinned substances in either the platform or substrate materials. Generally, these approaches are used to create atomic network structures under specific input conditions such as vacuum conditions, heat, laser energies, and pulse.

### 3. Surface and Structural Features of g-$C_3N_4$

Due to the very effective method in the domain of light-induced inorganic oxide semiconductors, the photocatalyst performance of g-CNs is seriously hampered. Pristine g-CNs' low specific surfaces and rapid photoinduced charge pairing allow a severe note to change its architecture and banding locations. The varied geometrical patterns paired with these g-CNs could be used to boost the quick recombination process, leading to increased photoelectrocatalytic or electromechanical activities during catalytic reactions. Nanocrystals in different sizes and shapes, including wires, cylindrical configurations, and nanotubes, are employed to modify the catalytic effect of hybridized g-CNs [9,37]. The core covered component of GCN facilitates improved results of GCN-based photocatalytic device applications to its pp-stacked heterocyclic thicker systems of heptazine (tri-s-triazine) or triazine heterocycles (Figure 1) [38].

**Figure 1.** Structural orientation of a g-$C_3N_4$ single layer. N atoms shifted under $\alpha$ and $\beta$ varieties of locations [38].

Important progress in the physicochemical aspects of composites such as carbon nanotubes, graphite, and other 2D configurations are made in the form of one- and two-carbon nitride films using appropriate processes [18,31]. Nanomaterials, g-CN, and nano-ranged thick polyheptazine sheets were commonly used to distinguish two metal-based semiconductors with suitable topologies. As a result, there is a huge requirement for exfoliated GCN and its counterparts to develop promptly in extended surfaces, which include 2D g-CN films and nanowires.

### 3.1. Structural and Morphological Aspects

The structure and morphology of the CN are often studied using a TEM or SEM approach that has been used to describe the usual membranous nanosheets with many mesoporous features. At some atmospheric conditions, polycondensation of urea is commonly used to create a similar atomic network structure of the CG. Pure CG frequently has nanosheet-like architectures; however, mixed CG nanocomposites have a variety of topologies based on the synthesis procedure. To explore the features of CN, foundation methods such as FTIR, XRD, HRTEM, SEM, XPS, and others have been developed [22,39]. Alternatively, theoretical methodologies could be used to elucidate the electronic structure, electrical properties, and surface characteristics of these conjugated polymers.

The interfacial and compositional content of the CN, as well as the atomic proportion of pristine and hybrid forms, should be investigated using XPS which is strongly advisable. As a result, a brief overview of the basic properties of CN is emphasized progressively inside this part, utilizing appropriate methodologies. The HRTEM has previously been shown to accurately characterize the structure, distribution, and scattering tendency of mixed or native CN. To examine these sheet-like objects, the regulated electron beam energy (in eV) has been tuned. The strong vacuum, full-beam power, and sample preparation inappropriate solutions all aid in confirming the surface morphology of the CN as well as its derivatives. HRTEM-based studies are commonly observed and could be used for core morphological features, and as a result, this approach has become an indispensable method for evaluating CN morphology. Wider research has been carried out to functionalize such 2D materials using chemical and physical approaches, and HRTEM, AFM, and SEM were used to examine their topology. FTIR is also used to examine the interfacial bonding of CN and cobalt sulfides (hexagonal sheet-like) contact. For the morphological studies, Guan Wu and workers reported layered or sheet-like structures for the g-$C_3N_4$ synthesized by different methods. TEM analysis were used to determine the morphology of these

layered sheets, shown in Figure 2a, respectively [40,41]. Researchers constructed g-C$_3$N$_4$ by employing a chemical route under the condensation of cyanamide as a molecular starting precursor. The formed sample revealed 2D oriented highly ordered sheet structures of g-C$_3$N$_4$ studied by SEM, shown in Figure 2b–d. However, the XRD profiles of the core g-C$_3$N$_4$ (JCPDS 87-1526) clarify the crystalline nature of 2D structures, Figure 2e. The (002) crystallographic index of g-C$_3$N$_4$ is related to the sharpest peaks for pure g-C$_3$N$_4$ at $2\theta = 27.4°$. The modest signal at around $2\theta = 13.1°$ relates to triazine units repeating in a plane [41]. This crystallographic plane is ideally considered to determine the solid nature of the synthesized graphitic carbon nitrides by several researchers. The layered structure under nano range shows the difference in intensity as compared to bulk g-C$_3$N$_4$ sheets.

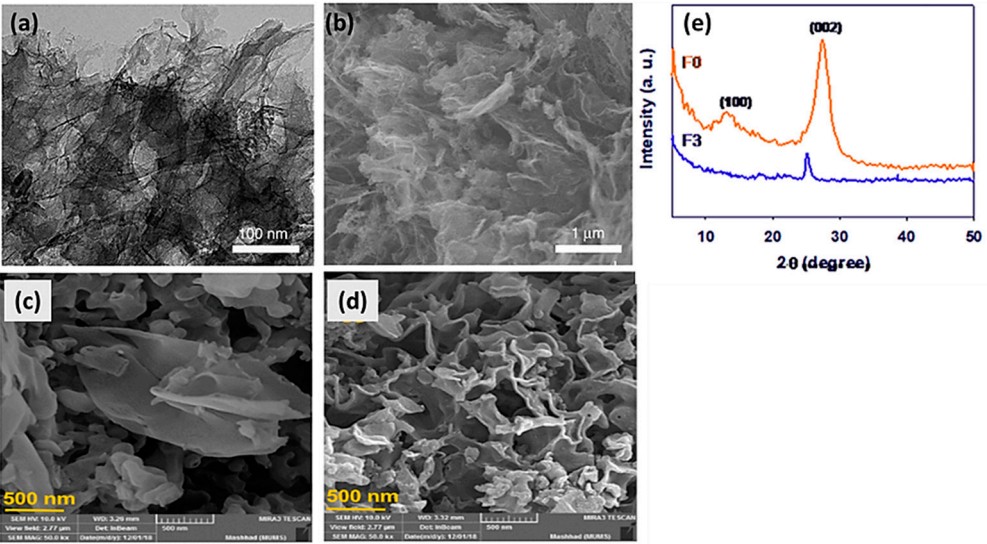

**Figure 2.** Nanoporous g-C$_3$N$_4$: (**a**) TEM images of g-C$_3$N$_4$ nanoporous; (**b**–**d**) SEM micrographs, (**e**) XRD profile of g-C$_3$N$_4$ [38].

### 3.2. Crystal Phase Confirmations

The selected area electron diffraction (SEAD) investigation is also quite significant. The diffracted pattern during TEM investigation could also validate the crystal form in terms of single or polycrystalline substances. Based on SAED results, the majority of the studies were deemed to be very legitimate [42,43]. Due to the lightweight atoms involved in carbon nitrites, diffraction patterns in 2D nanosheets are observed to be rare. Due to the distinct diffraction patterns under the SAED pattern in TEM research, heavy metals such as transitional, alkali, and lanthanides have been examined extensively. Understanding phase formations and characteristic d-values for essential sections are aided by the interplanar distance with diffracted spots. Furthermore, Runjia Lin [9] used the TEM to examine the layered frameworks of these 2D films and discovered a considerable distribution variation amongst functionalized carbon nitrite stacks with graphene oxides at different molar ratios. The homogeneity of the CN and graphene-based CN is influenced by surface-functionalized groups and solvent dispersion. Moreover, carbon–nitrides hybrids containing other conjugated polymers such as MoS$_2$, Co–sulfides, and Ni–sulfides were produced and characterized using certain essential methodologies including XRD, which was used to determine the contact between these two-layered materials [44,45].

With the use of crystallographic planes derived from high-resolution TEM, and crystallographic planes in XRD, the creation of heterojunctions between 2D alternating layers might be verified. The critical importance of these heterojunctions in enhancing work capabilities in energy storage devices has been explored and thoroughly explained. For this kind of carbon object, the FTIR method is also employed to detect functional active affinity. Such approaches were used to investigate the modes of vibration of CN and functionalized

CN (Figure 3a). The specific C–C and C–N stretched harmonics are unique to the 2D layers in which certain FTIR wavenumber intensities are found.

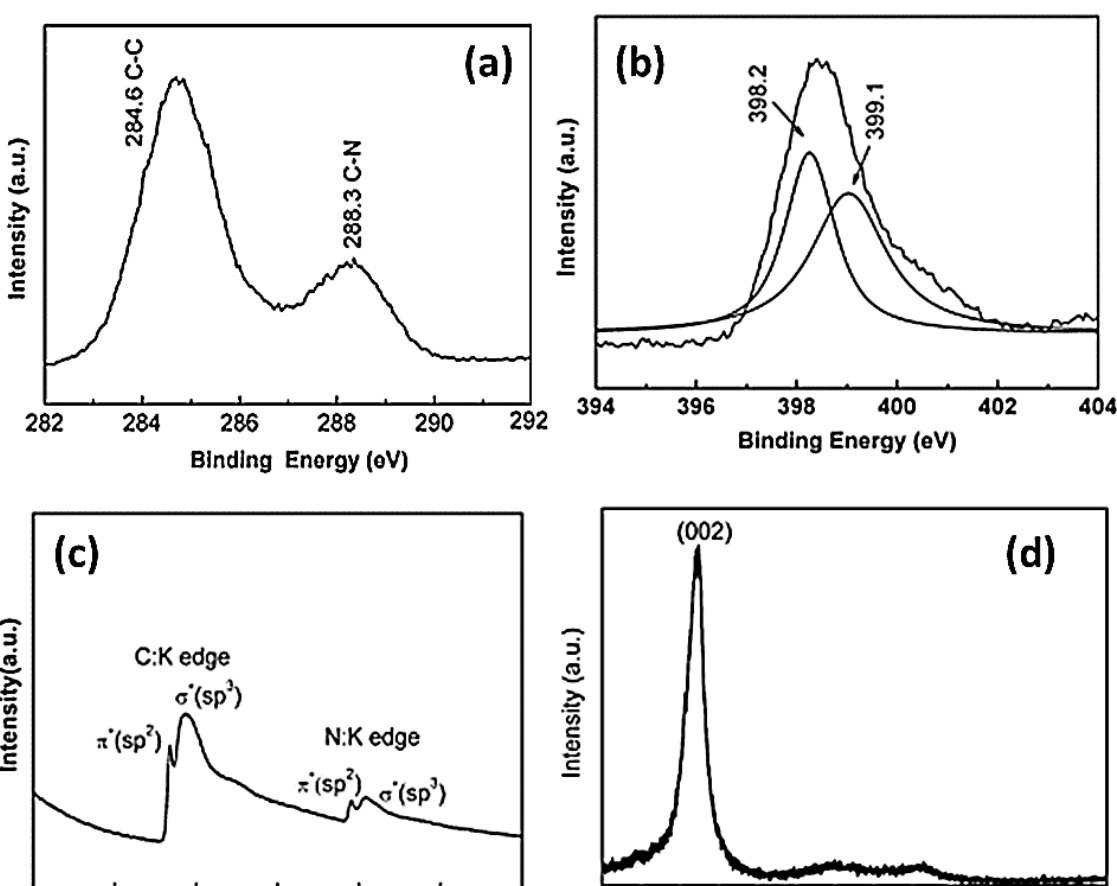

**Figure 3.** Elemental analysis of the g-$C_3N_4$; (**a**) C1s XPS peaks of g-$C_3N_4$; (**b**) N1s XPS peaks of g-$C_3N_4$; (**c**) electron energy loss spectroscopy spectrum; (**d**) a characteristic XRD spectra of g-$C_3N_4$ [46].

*3.3. Elemental Analysis: g-$C_3N_4$*

Because CN has a 2D character comparable to that of a thin coating, many processes are utilized to authenticate the layers, as previously indicated. Body, redox arrangements, and chemical ratios are all highly valued using the XPS method, which is one of the most reliable. XPS is utilized to analyze CN frameworks in this circumstance, with specialized analysis conditions such as super vacuum and X-ray frequency. The likely components of the CN may be easily discovered using scanning analysis, but for the functionalized assembly, extraordinary scanning for the individual parts is necessary (see Figure 3b–d).

## 4. Significance of g-$C_3N_4$ towards Energy Storage and Environmental Impacts

Due to its excellent adsorption capacities, excellent power transfer, and huge surface architecture, hybrid carbon nitride has aroused a lot of interest in the last decade. It also has a low resistance and is the strongest carbon nitride allotrope at ambient temperature due to its low resistivity and stability. As reactivity rates increase, organometallic carbide hybrids outperform catalytically active oxides in respect of thermomechanical endurance and oxidation stability. Processing, nanosensors, and rechargeable batteries could all benefit from the catalytically active carbide composition [17,32,33].

### 4.1. g-C₃N₄ for Energy Storage (Supercapacitor) Uses

Supercapacitors have sparked attention from scholars due to their high power efficiency, rapid charging rate, and excellent durability. An important research difficulty in the growth of advanced supercapacitors is the development of electro catalysts with a layered form, high electrochemical sensitivity, high surface to volume ratio, and a simple production procedure. Metal-loaded carbides are frequently regarded as 2D alternatives for building electrochemical technologies due to their exceptional oxidation–reduction and high conductivity [47,48]. Its applicability in realistic science and innovation is confined, however, because of its thermodynamically stable layered architectures and reduced reactive sites/area. Hybrid materials have the potential to effectively remedy defects in the changed surface and characteristics of single atomic requirement analysis while also optimizing their separate tools to increase the substance's positive wealth [49,50].

The g-C₃N₄ polymers have been discovered to be particularly useful in improving the operation of supercapacitors. Thus, Sharma Meenu et al. [51] conducted CV curves of pure metal oxides such as $ZnCo_2O_4$ and g-C₃N₄@$ZnCo_2O_4$ blended electrodes. The enhanced capacitance hits were performed at a scanning speed of $20\,mVs^{-1}$ from 0.0 to 0.5 V potential. The hybrid CN doped with $ZnCo_2O_4$ showed a greater current amplitude. The greatest power rating of the composite g-C₃N₄@$ZnCo_2O_4$ electrode is $157\,mAhg^{-1}$ at $4\,Ag^{-1}$. At $10\,Ag^{-1}$, the single electrode premised on g-C₃N₄@$ZnCo_2O_4$ has capacity retention of 90% after 2500 constant GCD cycles. g-C₃N₄@$ZnCo_2O_4$ materials surpass other carbon-based materials due to their exceptional optical properties, electrical conductivity, and thermally stable system. The insertion of nitrogen to g-C₃N₄ enhances the surface area and capacitance while maintaining the cyclability of the manufacturing and product. As a result, the g-C₃N₄@$ZnCo_2O_4$ electrode is incorporated into a hard surface apparatus for energy storage research. The electrochemical studies of casting-produced C3N4 electrodes were performed by Roger Gonçalves and co-authors [36] using cyclic voltammetry. A capacitance cost of $113.7\,F\,g^{-1}$ at $0.2\,A\,g^{-1}$ along with an 89.2% retention were obtained after 5000 charge and discharge cycles at $3.0\,A\,g^{-1}$ (Figure 4). The designed electrodes expressed a remarkable specific energy value $\approx 76.5\,W\,h\,kg^{-1}$ at $11.9\,W\,kg^{-1}$ operation power. These electrochemical findings are appreciated to lead to carbon nitride-based hybrid materials for improved energy-based applications in optoelectronic devices and sensors.

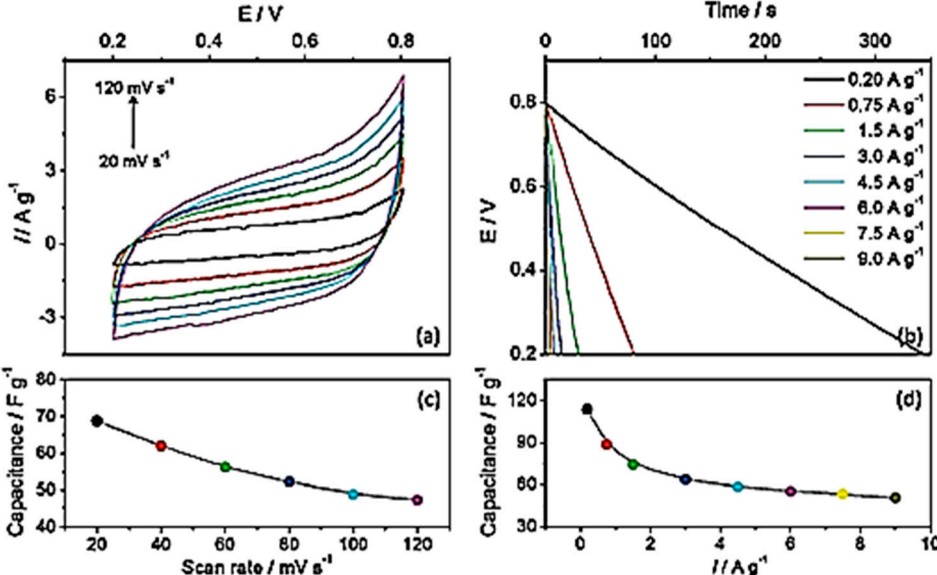

**Figure 4.** (**a**) Cyclic voltammetric curves at multiple scans, (**b**) discharge curves, and specific capacitance, (**c**) scan rates, (**d**) specific current [36].

According to Samarjeet Singh Siwal et al. [15], CuMnO₂–gCN exhibits impressive electrocapacitive behavior, with good conductivity ($817.85\,Ag^{-1}$) and deserving rate capability

(1000 cycles). Certain metallic elements, such as $MnO_2$ and CuO, are price solutions for transparency, sensitivity, and durability, to name a few further advantages, while $CuMnO_2$ has a high surface area and homogeneous distribution, enabling additional functional zones to be accessible. As a result, $CuMnO_2$–gCN was used as an effective electrochemically active catalyst to boost enhance efficiency, resulting in capacitance of 817.85 $Ag^{-1}$ after 1000 cycles (stability). It has also been discovered that metal-based metasurfaces, particularly modified metal-oxides, are commonly employed in these electrochemical areas because of their increased electron mobility and hence larger voltage density. Multiphase substances are therefore constructed and exploited to boost conductivities by modifying the electronic structure of metal-oxides through the inclusion of appropriate ions or metals into the electrode material. Transitional and lanthanide host materials, such as oxides and sulfides, have been discovered to be the most desirable composition because they prevent carrier recombination. During studies or measurements, the altered ions cause imperfections in the bulk and interface of the host nanocomposites, triggering electron transport. Kannadasan Thiagarajan and workers [52] employed a hydrothermal technique to produce $NiMoO_4$/g-$C_3N_4$ as an electrocatalyst material for electrochemical capacitors systems. The $NiMoO_4$/g-$C_3N_4$ composites displayed a higher highest capacitive performance than native $NiMoO_4$ (203 $Fg^{-1}$) (510 F $g^{-1}$). The $NiMoO_4$/g-$C_3N_4$ hybrid electrocatalyst was also incredibly reliable, with up to 91.8% specific capacity after 2000 charge–discharge cycles. Ultimately, it was discovered that the capacity factor of $NiMoO_4$/g-$C_3N_4$ is 11.3 $Whkg^{-1}$. These findings demonstrated that $NiMoO_4$/g-$C_3N_4$ would make an excellent reactive component for electrolytic capacitors. Veena Ragupathi and coworkers [14] created a hemispheric g-$C_3N_4$ mixed MnS substance by using the sol–gel technique and discovered a 463.32 F $g^{-1}$ reversible capacity with a 10 mV s1 scanning speed, with a 98.6 percent specific capacity after 2000 cycles (Figure 5). At 0.005 A $g^{-1}$ charge density, although, the g-CN functionalized MnS had a superior 403.36 F $g^{-1}$ maximum power ability.

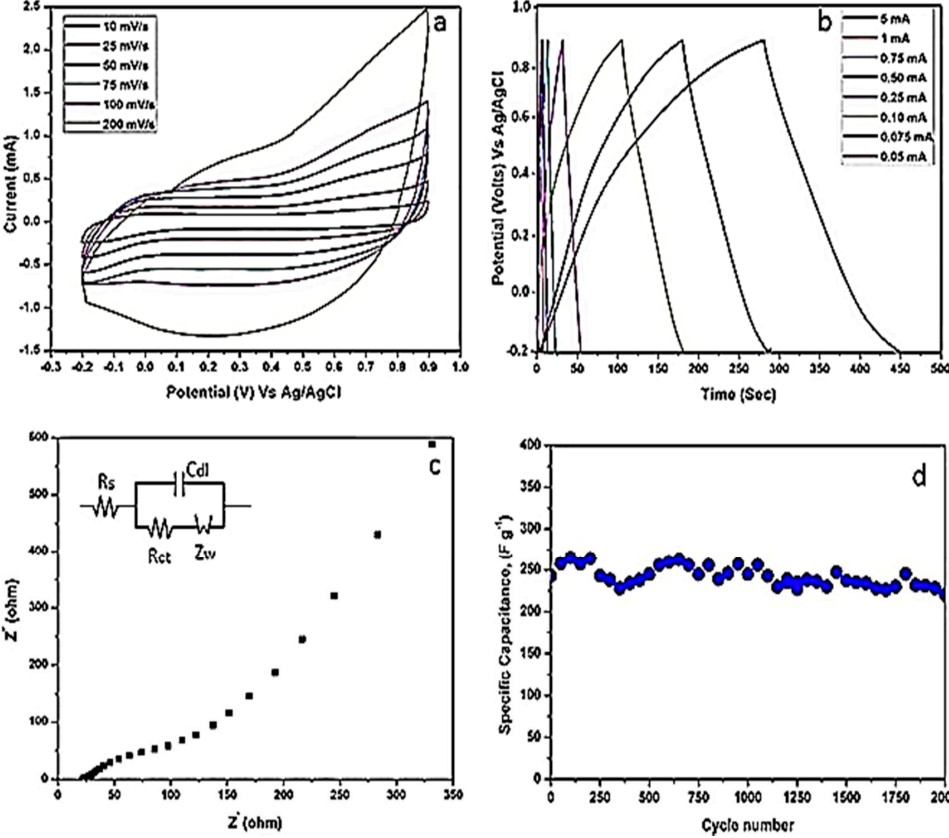

**Figure 5.** Electrochemical studies for g-CN loaded MnS: (**a**) CV plots, (**b**) galvanostatic charge and discharge plots, (**c**) Nyquist plot, and (**d**) plots for specific capacitance vs. cycle number [14].

Likewise, MnS acted as a positive electrode and an activated carbon-based negative electrode to create solid unsymmetrical circuits for the energy storage investigation, which showed 573.9 F g$^{-1}$ at 0.5 A g$^1$. The particular sphere and packing properties of g-C$_3$N$_4$ improve electron transport while also strengthening the material's internal structure. Thus, the architecturally formed 2D layers can be used as electrochemical devices to boost the supercapacitor's functional ability when loaded with inorganic metal-based oxides, or even sulfides. Minmin Liu et al. [22] used the pyrolysis procedure to create g-C$_3$N$_4$@Ni$_3$C 2D structures at ambient temperature. The supercapacitor performances of the developed g-C$_3$N$_4$@Ni$_3$C structures in the form of nanosheets have been excellent. The functional ability of such graft-conjugated polymers was also proved in hydrogen generation by NaBH$_4$ hydrolyzed, as well as a reduction in p–nitrophenol. This method presents a fresh method for the production of sustainable refined fuels from low-cost building ingredients. With respect to structure, the large interfacial sites and volume help increase electrical and chemical catalytic reduction activity. Graphene, activated carbon, C-aerogel, and CNTs are the most often used electrode elements in EDLCs [19,22,31], while additional metal-based frameworks are used to increase the chemical potencies in supercapacitor electrodes. Due to their unusual characteristics, such as high SC ratings, incredible performance/energy densities, and fast reversible electrochemical reactions at the conductive junctions, experts have been attracted to all these metal oxides.

As a result, a research study showed that rGO, GO, pyrrole, and polyaniline, as well as other conducting material chemicals, were employed to increase the conductance of MoS$_2$. Although showing tremendous progress, MoS$_2$ and some other compounds electrical and chemical reactivity continue to be inadequate. The compact MoS$_2$ sheets grafted on g-C$_3$N$_4$ have 2D/2D heterostructures geometries. Sufficient consistency, high surface area, outstanding electron–donor character, cheap, and environmentally friendly nature are some of the improved heterostructure qualities of graphitic carbon nitride [53]. Larger charge redistribution features on the interface coupled with more electrons onto the g-C$_3$N$_4$ boost the substance's conductance. Polymeric metasurfaces, which are essentially metal nanoparticles connected to layered materials, metal-oxides, and multimetallic frameworks, have been used in the field of science and technology in the past [54–56]. The compositional, spectroscopy, and interface alterations of these nano-ranged substances have revealed their beneficial spectrum and physical characteristics, approaching individual dynamics. While the durability and specific energy distribution of double-layered capacitive substances, such as carbon nanotubes, have been established and appreciated, the power output constraint has also drawn a lot of interest. When superior electron capacitive components, such as graphene, activated carbon, metal oxide, transition sulfide, or conducting polymers [57,58], are combined with carbon nanotube-oriented fibers, the conductivity of the native fibers can be greatly increased by maintaining the periodic features. In this context, Peng and colleagues [59] constructed a CNT/graphene composite fiber by covering the as-prepared MWNT assemblies with colloidal GO and spinning them into fibers. The transfer tendency of electrons in the composite fiber is noticeably enhanced because of contact of the pi–pi bond between the graphene oxide sheet and the carbon nanotube, and the graphene oxide layer can reduce carbon nanotube packing and enhance ion pathways. In comparison to naked CNT fiber (630 MPa), the chemical-reduced blended fiber seems to have a mechanical property of 500 MPa, whereas the conductance of the fiber reinforcement can be as strong as 450 20 S/cm. In comparison to the 5.83 F/g of the pure CNT fiber, the mixed fiber's computed specific capacity was 31.5 F/g, while Foroughi et al. [60] investigated a unique type of inductive carbon nanotube–graphene hybrid fiber under electrospinning chemical-reduced graphene within and covering the face of MWNT fiber during the drawing phase, based on a similar principle [61]. The functionalized fiber had electrical properties of 900 50 S/cm, whereas its impact resistance, stiffness, and modulus of rupture were all about 140 MPa, 2.58 GPa, and 6%, correspondingly. At a scanning speed of 2 mV/s, the specific capacity was considerably enhanced to 111 F/g. The co-spun CNT-based fiber could also improve the fundamental electrocatalytic activity as a comparison to the naked

CNT fiber with the use of nanocrystals or nanoflakes. As a result, such nanomaterials or equivalents have been currently widely used in the creation of sophisticated raw resources or components for nanoelectronics and allied equipment [25,62,63].

### 4.2. Environmental Applications

Scientists are very interested in these materials because of their outstanding electrical and chemical properties. Carbon nitride is found in nature in five different crystalline forms, according to research. Along with its chemical reactivity, mobility, and sensitivity, graphitic carbon nitride (g-$C_3N_4$) has received much interest [64,65]. The g-$C_3N_4$ is made up of tris-triazine subunits with a lot of extra electrons and has a planar-conjugated shape [46]. Due to the obvious chemical bonding formed between g-$C_3N_4$ and metal ions, it is applied to eliminate pollutants from wastewater. To interpret a hybrid iron-oxides nanocomposite ($Fe_3O_4$–g-$C_3N_4$), $Fe_3O_4$ nanostructures loaded on g-$C_3N_4$ were designed by Shuangzhen Guo et al. [46] under the ultrasonication method. They used SEM, X-ray diffraction, vibrating sample magnetometer (VSM), Fourier transform infrared, thermogravimetric analysis, and particle size analyzer to investigate the morphology and compositions of the formed catalysts, and the magnetic adsorbent $Fe_3O_4$–g-$C_3N_4$ was used to extract Cd(II), Pb(II), and Zn(II) from aqueous medium.

Adsorption in real samples: Experiments in electrolytic zinc residue percolate were used to study the adsorption ability of as-prepared $Fe_3O_4$–g-$C_3N_4$. The effluent was made up to conduct the adsorption studies by purifying it through a membrane filter. The obtained adsorption capacities for the $Fe_3O_4$–g-$C_3N_4$ nanocatalysts for Cd(II), Pb(II), and Zn(II) were 50.41, 69.38, and 82.31 mg/g, respectively (Figure 6).

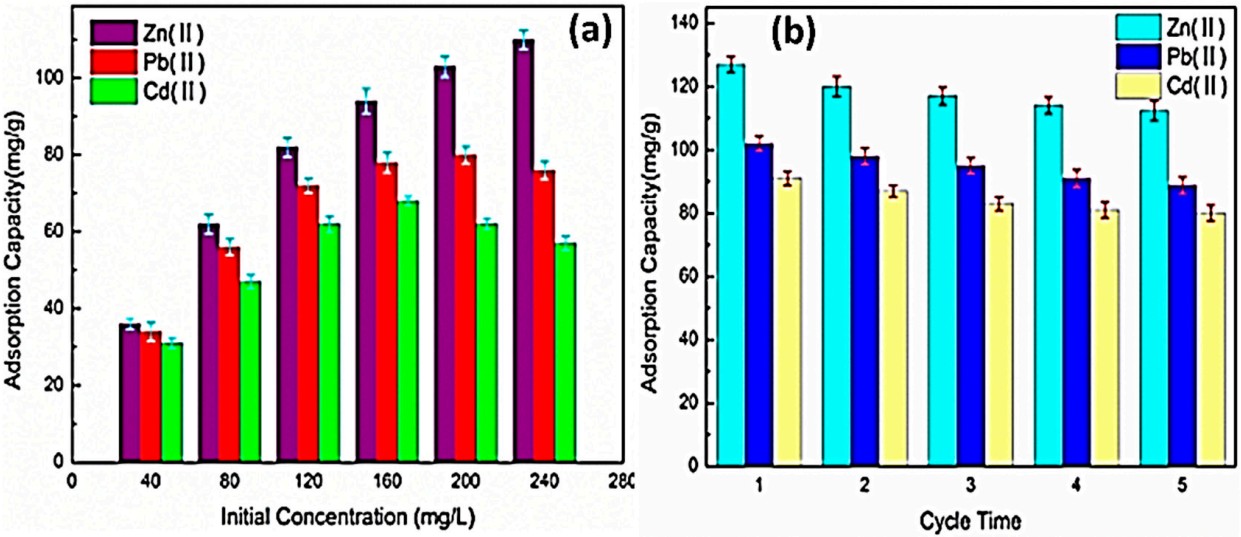

**Figure 6.** (**a**) Impact of metal competition on adsorption capacity (metal ion concentration, 200 mg/L), (**b**) cycle adsorption studies of Zn(II), Cd(II), and Pb(II) by $Fe_3O_4$–g-$C_3N_4$ [46].

However, melamine and industrial waste residue (red mud) were taken as source components in a one-step thermal polymerization process to create the red mud/g-$C_3N_4$ (RM–CN) composite [66]. Because of the synergistic effect of adsorption and photocatalysis, RM–CN composites have a massive impact on the removal of organic contaminants from sewage. The surface morphology of RM–CN was considerably enlarged, concerning only CN, due to the inclusion of RM. The refined 0.8 percent RM–CN product (with RM mass concentration of 0.8 wt% in precursor) exhibited considerable photodegradation effectiveness for antibiotics and organic molecules with excellent recycling ability when exposed under visible spectrum (Figure 7).

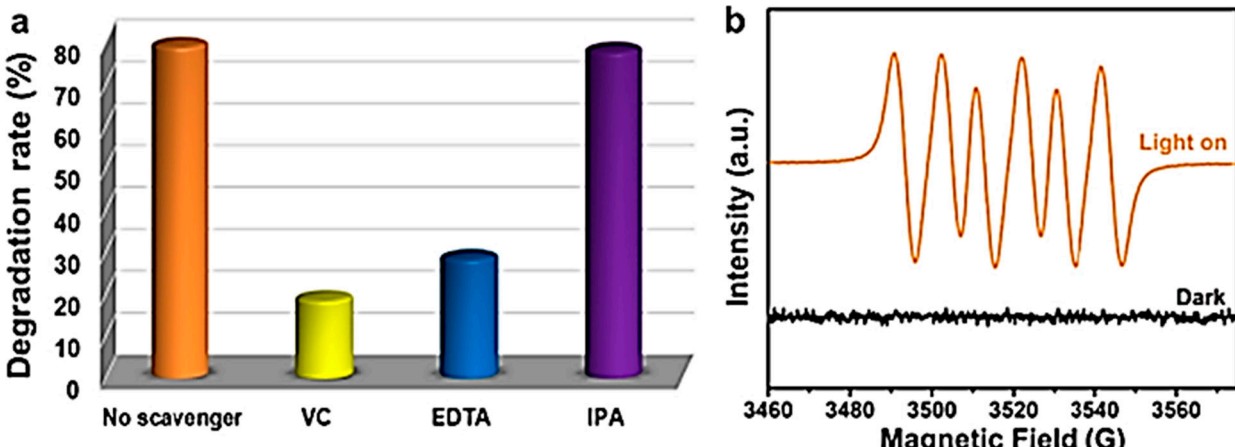

**Figure 7.** (**a**) Trapping studies of active moieties throughout photocatalytic degradation of TC over 0.8% RM–CN. (**b**) ESR bands of % $O_2^-$ trapped by DMPO over 0.8% RM–CN sample in dark and light irradiation [66]. Cu(I)-loaded conjugated carbon nitride structures (Cu–CNF) along with the heteroatom N and O were structured followed by the ligand charge transfer [67]. Cu–CNF catalysts were employed efficiently to catalyze chlortetracycline hydrochloride chemicals from the deionized water. The elimination rate was high in the case of the river water (68.2%) corresponding to the tap water (45.7%), and swine wastewater (45.7%) (33.1 percent). For the improved catalytic personality of g-CN materials, Xiaohu Zhang and co-workers [68] reported fundamental characteristics by discovering a strong asymmetric phthalocyanine (sensitizer) of graphitic carbon nitride. They used it as efficient photocatalysts material for the production of $H_2$ on the near-IR spectrum. The zinc phthalocyanine derivative (Zn-tri-PcNc) associated with the chenodeoxycholic acid (CDCA) showed a 125.2 μmol h$^{-1}$ efficiency due to an increased recombination rate. It was found that the photoactivity for $H_2$ is increased by the activated Zn-tri-PcNc dispersed on the 2D g-$C_3N_4$ surface and photo-induced free electrons inserted into the CB of g-$C_3N_4$, and is captured by supported co-catalyst Pt for $H_2$ generation on water reduction (Figure 8). Concurrently, the oxidized Zn-tri-PcNc is produced via accepting an electron by the reagents which further excites under illumination. The photostability for $H_2$ creation from the Zn-tri-PcNc/g-$C_3N_4$ system was traveled on 700 nm monochromatic radiation.

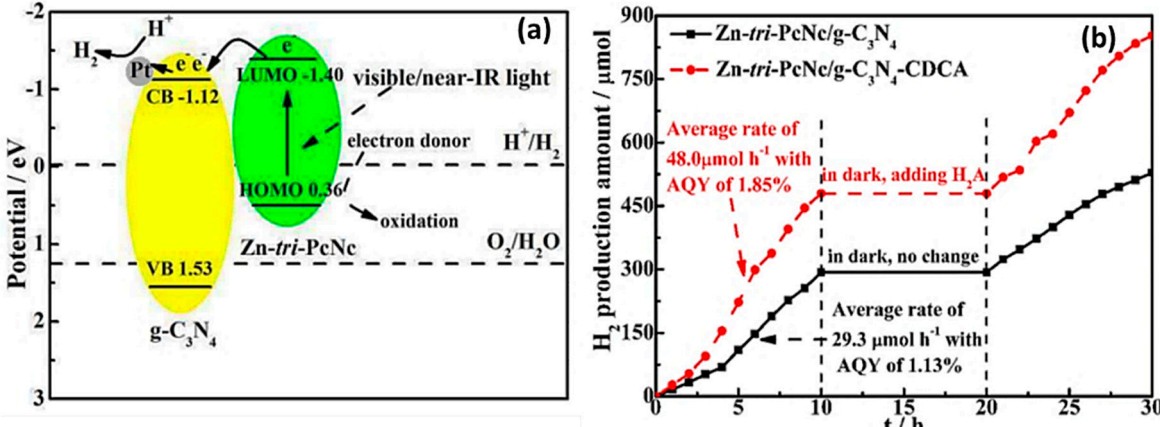

**Figure 8.** Photocatalytic-based $H_2$ production process (**a**) and stability for $H_2$ production (**b**) of Zn-tri-PcNc/g-$C_3N_4$ materials.

Therefore, these are some basic studies performed for wastewater learning to protect environmental health. Additionally, numerous nanomaterials and their functionalized phases were studied for their environmental impacts and biomedical significance due to their modified surface, shape, sizes, and band structures [28,62,63]. In the present article, wider information of carbon nitride-based materials is focused on including manufacturing and their applications in medical science and environmental technologies [69–71].

## 5. Summary and Conclusions

The mutual understanding of carbon nitride-based heterostructures for energy storage and conversion is stated in this report. The features of each stage are highlighted and explored regarding the bigger group of g-C$_3$N$_4$ substances. Conversely, two-dimensional polymers such as carbon nitride, graphene, and CNTs materials have been regarded as viable candidates, although this article focuses mostly on carbon nitride. Synthetic processes and elements in the process are used to develop extraordinary physical and chemical strengths and electrocatalytic activity. Moreover, these conducting polymers have found their way into a variety of uses, mostly in power systems, and so this chapter provides a systematic review. To improve the performance of electrochemical devices, a basic investigation of conductive polymers associated with their combinations was examined. Comparatively, the prospective uses of carbon nitrides, as well as their combinations, are explored, exhibiting improved energy storage capacity. Certain principles were also explored by interface alterations, electronic structures, and compositional analysis. Because of current efficiency, two-dimensional frameworks such as graphene, CNTs, and the modified g-C$_3$N$_4$ are recommended for supercapacitance usefulness. The relevance of two-dimensional carbon nitride is explored and expanded to investigate these substances as real and future carbon allotropes for supercapacitors applications. In conclusion, the subject of g-CN appears to be well worth further investigation, and it is expected to yield innovative metamaterials in the same way that related topics such as CNT structures and carbon mixed arrays have. We anticipate that our study and findings will ignite attention in g-CN as a promising advanced substance and will contribute to more progress throughout this intriguing subject.

**Author Contributions:** Conceptualization, G.K.I. and V.K.Y.; data curation, S.B.V., S.D.K. and G.K.I.; methodology, V.K.Y., G.K.I., S.D.K., M.K.S. and I.H.A.; validation, K.K.Y., I.H.A., Y.A. and S.Y.; formal analysis, G.K.I., B.-H.J., S.I. and Y.A.; resources, K.K.Y., B.-H.J. and S.Y.; writing—original draft preparation, G.K.I., V.K.Y., S.D.K. and M.K.S.; writing—review and editing, G.K.I., S.D.K., V.K.Y., S.I., K.K.Y. and I.H.A.; supervision, G.K.I., K.K.Y. and V.K.Y.; project administration, I.H.A. and B.-H.J.; funding acquisition, I.H.A.; investigation, S.D.K., V.K.Y., S.Y. and S.B.V.; software, Y.A., S.I., M.K.S., G.K.I. and K.K.Y.; visualization, V.K.Y., B.-H.J., I.H.A. and Y.A. All authors have read and agreed to the published version of the manuscript.

**Funding:** The authors extend their appreciation to the Deanship of Scientific Research at King Khalid University for funding this work through the research groups program under grant number R.G.P. 1/105/42. This work was supported by the Mid-Career Research Program [grant number 2020R1A2C3004237] of the National Research Foundation of the Republic of Korea.

**Institutional Review Board Statement:** Not applicable.

**Informed Consent Statement:** Not applicable.

**Data Availability Statement:** Not applicable.

**Acknowledgments:** Authors are thankful to the Department of Chemistry, HVHP Institute of Post Graduate Studies and Research, Sarva Vishwavidyalaya, Kadi, Gujarat, 382715, Gujarat, India for providing the technical support.

**Conflicts of Interest:** The authors declare that there are no conflict of interest.

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
