# Peer review of "2D Personality of Multifunctional Carbon Nitrides towards Enhanced Catalytic Performance in Energy Storage and Remediation"

_applsci, doi:10.3390/app12083753_

Round 1
Reviewer 1 Report
The authors conducted a systematic overview of the 2D personality of multifunctional carbon nitrides towards enhanced catalytic performance in energy storage and remediation, which is an interesting topic. Overall, the manuscript is well organized. However, the following comments must be addressed before it can be considered to be published:
(1) Several spelling/grammatical errors:
---Abstract, page 1 line 13, “two – dimensional” should be “two-dimensional”
---Abstract, page 1 line 17, carbon nitrides’ ?
---Introduction, page 3 line 105, “its derivatives are been” ?
(2) This paper needs a review for grammatical consistency and, coherence.
(3) Why are figures 5 and 6 highlighted?
(4) It is recommended to provide more detailed information on the structural and morphological features of g-C3N4, which is important. Figure 2 is not clear enough.
(5) Why does g-C3N4 is a promising material for energy storage and conversion? Are there any applications or test projects? The authors should elaborate more on this.
(6) For a review paper, the number of references seems not enough for a comprehensive investigation.
Author Response
The comments file is attached, please find the attachments.

Reviewer 2 Report
The review is interesting and complete the state of the art in domain, however, in my opinion there are two major issues that should be resolved before publication.
A much more attention should be devoted to the obtaining methods.
Therefore, it will be of interest to extend the chapter referring to the synthesis of carbon nitrides. A good review is the following:
Carbon nitrides: synthesis and characterization of a new class of functional materials
S. Miller, A. Belen Jorge, T. M. Suter, A. Sella , F. Corà and P. F. McMillan
Phys. Chem. Chem. Phys., 2017, 19, 15613-15638, DOI: 10.1039/C7CP02711G
Another issue is related to applications. Thus, environmental applications are summary presented, and nanocomposites with excellent properties of photocatalysis are omitted, even they are important. Such examples are nanocomposites with metal phthalocyanines, as those presented in the following article:
Highly Asymmetric Phthalocyanine as a Sensitizer of Graphitic Carbon Nitride for Extremely Efficient Photocatalytic H2 Production under Near-Infrared Light
Xiaohu Zhang , Lijuan Yu , Chuansheng Zhuang , Tianyou Peng , Renjie Li , and Xingguo Li
ACS Catal. 2014, 4, 1, 162–170, https://doi.org/10.1021/cs400863c
Author Response
Dear Reviewer,
We have revised the manuscript as per the reviewer's suggestions. The comments file is attached and answers are given point-by-point. Please find the attached file.
Regards

Reviewer 3 Report
Inwati and al review the use of 2D multifunctional carbon nitrides for energy storage and other applications. While the subject can indeed benefit from a systematic review of the literature, the current manuscript lacks scholarly style and clarity. The review is based on 40 publications, which considering the broad and general title of the paper is rather limited. The prose is verbatim with several repetitions and unsubstantiated statements. For example, the authors do not cite any publication for the first 49 lines, and the entire introduction section that spans for over 100 lines cites no more than 17 publications. Further, the narrative in different sections is incoherent at best and is full of diversions away from the theme of the section. In summary, while a review on 2D multifunctional carbon nitrides is very much needed, the current manuscript does not do justice to this demand. In the opinion of this reviewer therefore, this work is not suitable for publication in the journal 'Applied Sciences'.
Author Response
We have revised the manuscript with solving the suggested points. Please find the attached file for the response.
Thanks

Round 2
Reviewer 1 Report
The authors have tried their best to answer the comments. The reviewer would recommend that the manuscripts be accepted for publication in Applied Sciences.
Author Response
The authors are highly thankful to the reviewer for the positive response (Acceptance).
Reviewer 2 Report
The article is ready for publishing.
Author Response
The authors are thankful to the Reviewer for giving a positive response for accepting the article.
Reviewer 3 Report
In this revision the authors have addressed comments made by the reviewers in the first round of review. They have also added some references related to the theme of the review. Although considerable changes have been made, the review is still lacks coherence and reads more like a collection of short description of disparate works. I suggest the authors undertake an independent reading of their manuscript and check errors regarding language and presentation. Once the authors do this, the manuscript can be accepted for publication. No further review is necessary.
Author Response
We are thankful to the reviewer for these minor suggestions. we read the language and revised it for a good presentation. Now we hope that our revised version would be more suitable for acceptance.
Regards